# Localisation of nitrate-reducing and highly abundant microbial communities in the oral cavity

**Joanna E. L'Heureux**[1]*, **Mark van der Giezen**[2,3,4], **Paul G. Winyard**[1], **Andrew M. Jones**[1], **Anni Vanhatalo**[1]

**1** University of Exeter Medical School, University of Exeter, Exeter, United Kingdom, **2** Department of Chemistry, Bioscience and Environmental Engineering, University of Stavanger, Stavanger, Norway, **3** Biosciences, University of Exeter, Exeter, United Kingdom, **4** Research Department, Stavanger University Hospital, Stavanger, Norway

* j.lheureux3@exeter.ac.uk

**Data Availability Statement:** All relevant data are within the paper and its Supporting Information files.

## Abstract

The nitrate ($NO_3^-$) reducing bacteria resident in the oral cavity have been implicated as key mediators of nitric oxide (NO) homeostasis and human health. $NO_3^-$-reducing oral bacteria reduce inorganic dietary $NO_3^-$ to nitrite ($NO_2^-$) via the $NO_3^-$-$NO_2^-$-NO pathway. Studies of oral $NO_3^-$-reducing bacteria have typically sampled from either the tongue surface or saliva. The aim of this study was to assess whether other areas in the mouth could contain a physiologically relevant abundance of $NO_3^-$ reducing bacteria, which may be important for sampling in clinical studies. The bacterial composition of seven oral sample types from 300 individuals were compared using a meta-analysis of the Human Microbiome Project data. This analysis revealed significant differences in the proportions of 20 well-established oral bacteria and highly abundant $NO_3^-$-reducing bacteria across each oral site. The genera included *Actinomyces*, *Brevibacillus*, *Campylobacter*, *Capnocytophaga*, *Corynebacterium*, *Eikenella*, *Fusobacterium*, *Granulicatella*, *Haemophilus*, *Leptotrichia*, *Microbacterium*, *Neisseria*, *Porphyromonas*, *Prevotella*, *Propionibacterium*, *Rothia*, *Selenomonas*, *Staphylococcus*, *Streptococcus* and *Veillonella*. The highest proportion of $NO_3^-$-reducing bacteria was observed in saliva, where eight of the bacterial genera were found in higher proportion than on the tongue dorsum, whilst the lowest proportions were found in the hard oral surfaces. Saliva also demonstrated higher intra-individual variability and bacterial diversity. This study provides new information on where samples should be taken in the oral cavity to assess the abundance of $NO_3^-$-reducing bacteria. Taking saliva samples may benefit physiological studies, as saliva contained the highest abundance of $NO_3^-$ reducing bacteria and is less invasive than other sampling methods. These results inform future studies coupling oral $NO_3^-$-reducing bacteria research with physiological outcomes affecting human health.

**Funding:** The author(s) received no specific funding for this work.

**Competing interests:** The authors have declared that no competing interests exist.

## Introduction

Advances in sequencing technology have allowed characterisation and in-depth analysis of the bacteria of the human oral cavity, with over 700 bacterial species having been identified [1, 2]. It has been possible to elucidate the communities of bacteria involved in oral nitrate ($NO_3^-$) and nitrite ($NO_2^-$) reduction. In the human $NO_3^-$-$NO_2^-$—nitric oxide (NO) pathway, the ingested dietary $NO_3^-$ is absorbed and enters the circulatory system [3]. The $NO_3^-$ is actively taken up by the salivary glands [4] and $NO_3^-$ is concentrated in the saliva [5]. Commensal bacteria reduce the $NO_3^-$ to $NO_2^-$ which is subsequently swallowed, enters the bloodstream, and may be reduced to NO under anoxic or hypoxic conditions [5]. NO produced via the $NO_3^-$-$NO_2^-$-NO pathway is important for a wide array of physiological processes [3]. Better understanding of the process of bacteria-mediated dietary $NO_3^-$-reduction may aid the optimisation of $NO_3^-$ administration for improving blood pressure regulation, cognition, protection against ischaemia-reperfusion injury and exercise performance [5–9].

The Human Microbiome Project (HMP) under National Institutes of Health (NIH) contains bacterial samples taken from many human sites, ranging from the oral cavity to the gut and skin microflora [10, 11]. Through use of the HMP, bacterial composition and abundance can be compared between body sites on a larger scale, which may not be practical in conventional clinical or physiological study settings. Therefore, analysis of the data generated by the HMP can be used to inform and direct smaller-scale studies.

The oral cavity comprises various bacterial niches, such as the teeth, hard and soft palates, gingival sulcus, cheek, lip and tongue dorsum [12]. Previous work has suggested that the highest reduction of $NO_3^-$ occurs on the tongue surface, which has been associated with the presence of $NO_3^-$-reducing oral bacteria [13]. However, whilst previous studies have determined some oral niche areas of bacterial colonisation, human sample sizes are typically low, with less than 20 study participants being common [1, 13–16].

The new availability of data from samples taken on a larger scale can enable a wider perspective on where oral bacteria colonise particular oral niches. Methods for measuring oral $NO_3^-$-reducing bacteria composition conventionally involve sampling from either the saliva or tongue surface [16–19]. The genera indicated by previous research to include potent $NO_3^-$ reducing species included *Actinomyces, Brevibacillus, Campylobacter, Capnocytophaga, Corynebacterium, Eikenella, Fusobacterium, Granulicatella, Haemophilus, Leptotrichia, Microbacterium, Neisseria, Porphyromonas, Prevotella, Propionibacterium, Rothia, Selenomonas, Staphylococcus, Streptococcus* and *Veillonella* [13, 16]. Of these, Doel et al. identified species within *Actinomyces, Rothia* and *Veillonella* as the most potent $NO_3^-$ reducers [13].

Saliva samples may be collected due to ease of sampling, whereas samples taken from the tongue dorsum may have more physiological relevance because $NO_3^-$-reduction is known to occur on the tongue surface [13]. However, samples taken from the tongue dorsum using a buccal brush may not provide a high yield of $NO_3^-$-reducing bacteria [authors' personal observations]. The aim of this study was, therefore, to determine the site(s) in the mouth where the highest abundance of $NO_3^-$-reducing bacteria reside in a larger human population, such as those found in the HMP, in order to inform methodology for future studies investigating the relationships between oral $NO_3^-$-reducing bacteria and human health.

## Materials and methods

### Data mining

Complete 16S operational taxonomic unit (OTU) tabulated data and sample mapping files were mined from the publicly available online NIH Human Microbiome Project Metagenomic

16S Sequence QIIME community profiling database [10, 11]. The HMP is a community resource project that provides free access to use the human microbiome data by the scientific community. The data are available to use under the original HMP ethical approval and informed consent (https://www.hmpdacc.org/hmp/resources/tools_protocols.php). The original data was obtained by the HMP Microbiome Project [10, 11], and are publicly accessible for download at: https://www.hmpdacc.org/HMQCP/. The authors did not have any special access of request privileges. Authors did not have access to participant identifying information and all samples were anonymous. The participants were healthy 18- to 40-yr-old adults and the exclusion criteria included the presence of systemic diseases including hypertension, cancer, immunodeficiency or autoimmune disorders, use of potential immunomodulators, and recent use of antibiotics or probiotics [20]. The OTU data contained participant sample identifiers, bacteria names, class and number of observations (OTUs). Sample identifiers and body sampling site were extracted from the HMP mapping file, where saliva and six oral sites were further extracted. Anonymised sample identifiers were used to match bacterial observations and oral sampling site. Data from the HMP included samples taken following completion of informed consent procedure from a cohort of 300 participants, where samples were taken at multiple sites [10, 11]. Original methods have been described by the HMP Consortium [11]. The 2012 HMP data were downloaded for the purposes of the present study on 15th of November 2016.

Genus level classifications were mined from this dataset. Seven oral sample types and a total of 1288 samples were analysed from the oral cavity. The oral sites included attached keratinized gingiva (n = 183, 96 males, 87 females), buccal mucosa (n = 186, 96 males, 90 females), hard palate (n = 183, 95 males, 88 females), saliva (n = 166, 90 males, 76 females), subgingival plaque (n = 188, 97 males, 91 females), supragingival plaque (n = 192, 99 males, 93 females), and tongue dorsum (n = 190, 98 males, 92 females). Before proceeding with analysis, we compared the $NO_3^-$ reducing bacteria between male and female subjects at each oral site. No significant differences were found between males and females within each oral site ($P>0.05$ for all comparisons). Consequently, we grouped the male and female samples together for comparison of $NO_3^-$ reducing bacteria between oral sites.

## Data analysis

Data downloaded from the HMP repository were processed and analysed using R statistical software [21]. Sample uniformity within each oral site, and between sites were visualised using multidimensional scaling (MDS) plots with the *edgeR* package [22]. Alpha diversity analyses were completed using *vegan* [23]. Median proportions were used to calculate Shannon Diversity Index and Simpson's Diversity Index for visualization of diversity across the seven oral sample types, and the Chao 1 Index was employed to estimate species richness.

Using the HMP data, the *Metacoder* R package was used to generate an in-built taxmap object, which returned 985 phyla at 45383 observations [24]. For the detection of differences between bacterial abundances of each of the seven oral sample types specified, *Metacoder* was used to detect pairwise differences in $\log_2$-ratio of median proportions [24]. For each oral site sample, the differences in read proportions were extracted for analysis of 20 $NO_3^-$-reducing bacteria of interest. For visualisation of bacterial differences in $\log_2$-ratio of median proportions across sites, heat trees were generated using *Metacoder* [24]. The percentage median relative abundance of genera and the corresponding standard error of the median were calculated to determine a list of the most relative abundant genera in each oral site. The most frequently occurring genera with the highest percentage relative abundance for each site were selected for direct comparison of the ten most abundant genera. The percentage median relative abundances of the remaining genera were aggregated to create the "Other" genera list.

## Statistical analysis

Both Adonis and MRPP were used to identify any statistical differences in Shannon Diversity Index, and Simpson's Diversity Index across the oral sites. The Wilcoxon signed-rank test was then applied find oral site-specific differences in diversity. For pairwise comparison between each oral site, statistical differences were computed using the Wilcoxon signed-rank test in R, due to the repeated sampling of each body site for each participant within the HMP database. Benjamini and Hochberg False discovery rate (FDR) correction was applied to control for multiple comparison testing and $p < 0.05$ was considered statistically significant.

## Results

### Analysis of alpha diversities

An overview of the bacterial composition of saliva and six oral sites revealed a high abundance of five major phyla: Firmicutes (44%), Actinobacteria (20%), Bacteroidetes (16%), Proteobacteria (15%) and Fusobacteria (4%). Shannon Diversity Index (H') and Simpson's Diversity Index showed that there were significant differences in microbial diversity across each oral site: Multiple Response Permutation Procedure (MRPP), $p < 0.001$; Permutational Multivariate Analysis of Variance Using Distance Matrices (ADONIS), $p < 0.001$. Chao1 indicated similar species richness (*Fig 1*). However, further testing showed that there were no significant differences in diversity between the saliva and subgingival plaque sites ($p = 0.08$), the saliva and supragingival plaque ($p = 0.17$) and the subgingival and supragingival plaque sites ($p = 0.73$).

### Pairwise comparison of $NO_3^-$ reducing bacteria $\log_2$-ratio of median proportions at each oral site

In saliva and six oral sites, the $\log_2$-ratio of median proportions of $NO_3^-$-reducing bacteria were compared at different taxonomic levels. Significant differences were observed across phylum, class, order, family and genus (*Fig 2*).

The well-established $NO_3^-$-reducing genera selected for comparison in the present study were *Actinomyces, Brevibacillus, Campylobacter, Capnocytophaga, Corynebacterium, Eikenella, Fusobacterium, Granulicatella, Haemophilus, Leptotrichia, Microbacterium, Neisseria, Porphyromonas, Prevotella, Propionibacterium, Rothia, Selenomonas, Staphylococcus, Streptococcus* and *Veillonella*.

Of the selected $NO_3^-$-reducing bacteria, pairwise comparisons of the subgingival and supragingival plaque sites exhibited the least differences in proportions of $NO_3^-$-reducing bacteria. The genera with the most significant differences in proportions observed in all oral sites included *Corynebacterium, Propionibacterium* and *Eikenella*. There were no significant differences in the proportions found in *Brevibacillus, Staphylococcus,* and *Microbacterium* in any of the pairwise comparisons.

Eight of the well-established genera including $NO_3^-$-reducers were found in higher proportion in the saliva than on the tongue dorsum ($p < 0.01$). These included *Campylobacter, Capnocytophaga, Corynebacterium, Eikenella, Porphyromonas, Prevotella, Propionibacterium,* and *Selenomonas*. At the tongue dorsum site, there was a higher proportion of *Actinomyces, Granulicatella, Neisseria, Rothia,* and *Streptococcus*. No differences in proportions were found in *Brevibacillus, Fusobacterium, Haemophilus, Leptotrichia, Microbacterium, Staphyloccocus,* and *Veillonella*. In each of the site comparisons and each of the 20 selected genera comparisons, saliva had the highest significant proportions of $NO_3^-$-reducing genera (*Fig 2 and Table 1*). Supragingival plaque had a significantly higher proportion of selected $NO_3^-$-reducing genera compared to subgingival plaque, including *Actinomyces, Capnocytophaga, Corynebacterium, Haemophilus, Neisseria, Rothia* and *Streptococcus* (*Table 1*).

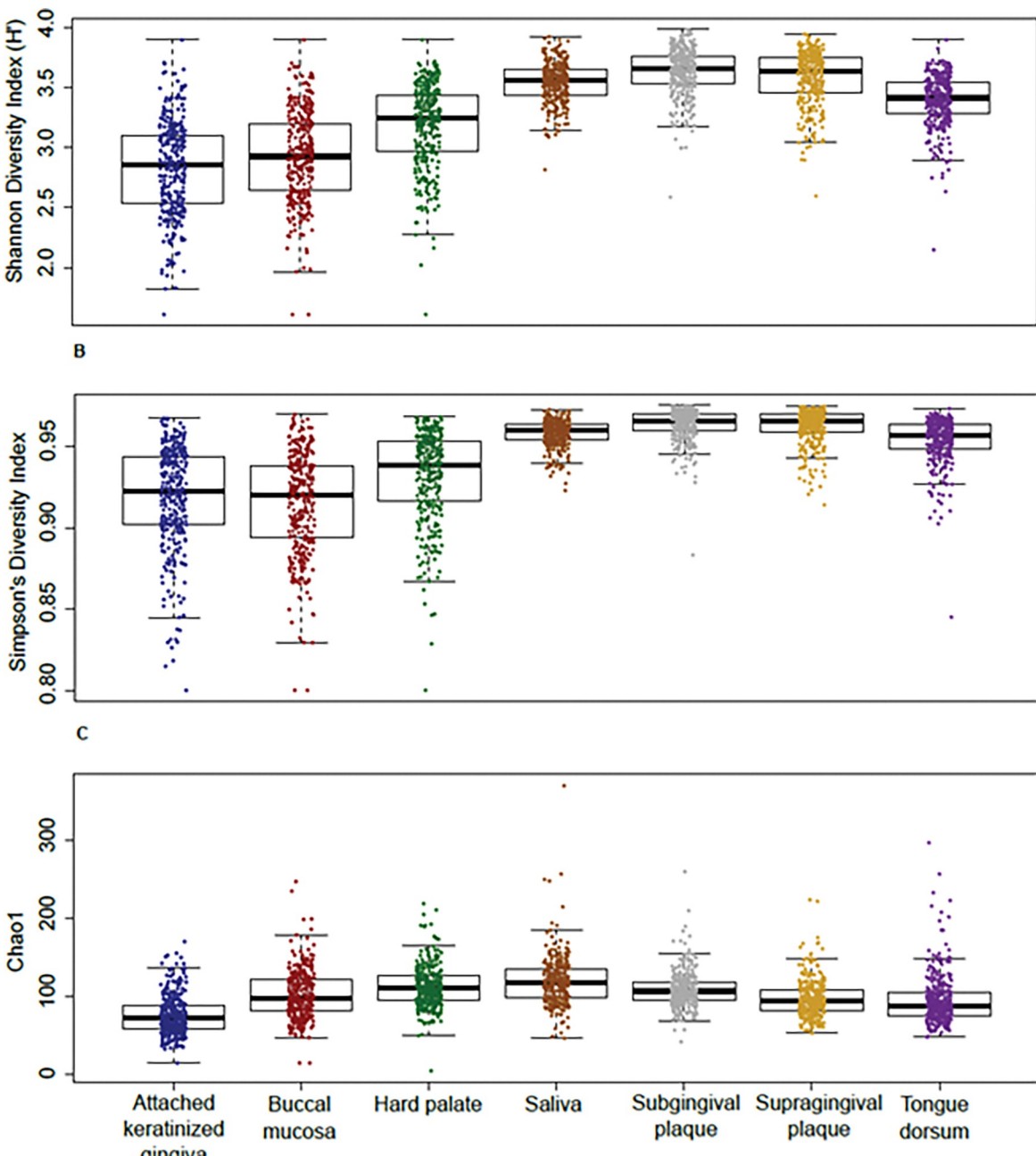

**Fig 1. Bacterial diversity in the oral cavity. A** Shannon Diversity (H') Index of saliva and six oral sites. **B** Simpsons Diversity Index. **C** Chao1 species richness for saliva and six oral sites. For each panel, scatter plots show the individual samples. The box plots show the median and quartiles.

Values show the differences in $\log_2$-ratio of median proportions between each oral site. Each pairwise oral site comparison is shown, where the oral site first stated is compared to the second oral site stated. Positive values are the taxa which were enriched in the first oral site stated, whilst negative values are the taxa which were enriched in the second site stated. For example, in the attached keratinized gingiva and buccal mucosa pairwise comparison, *Actinomyces* was enriched in the buccal mucosa site, whilst *Haemophilus* was enriched in the attached keratinized gingiva site.

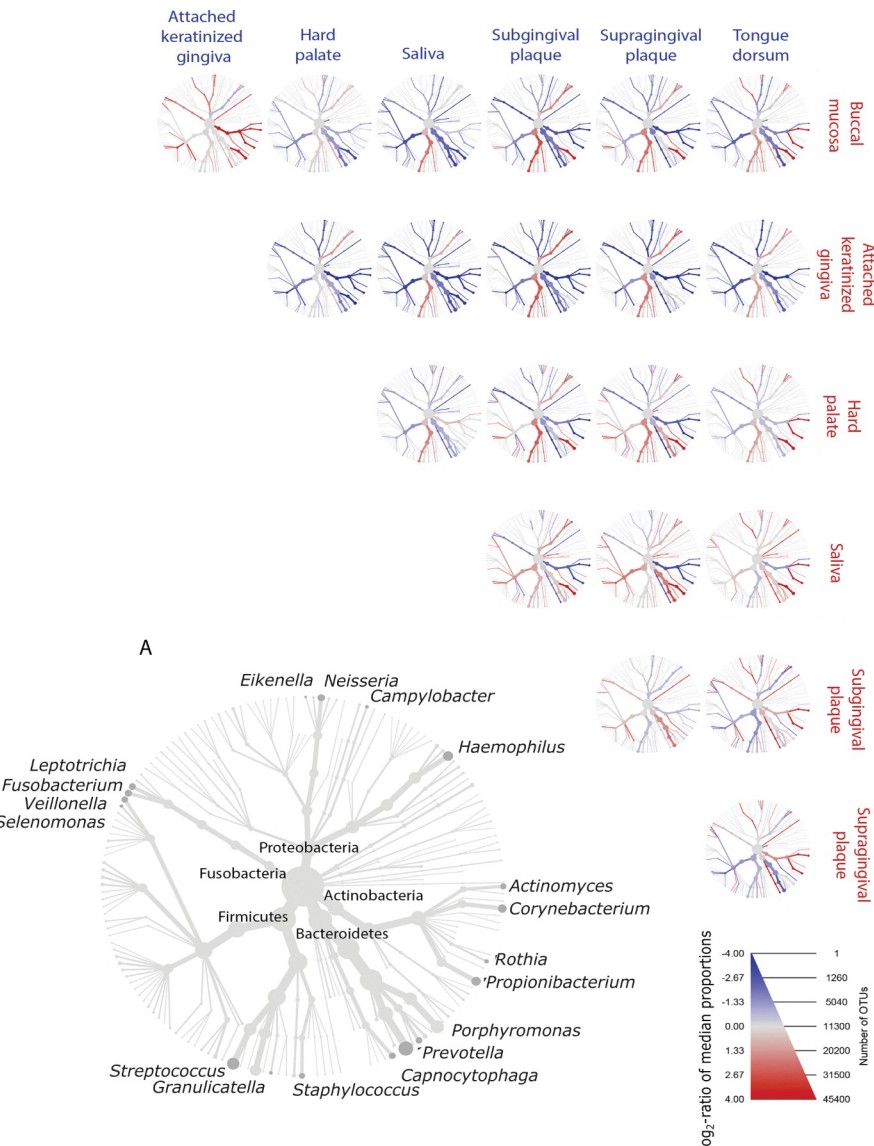

**Fig 2. Heat trees of pairwise comparisons showing the difference in median proportions of NO3⁻-reducing bacteria between seven oral sample types.** The heat trees show statistically significant differences in $\log_2$-ratio of median proportions of $NO_3^-$-reducing bacteria at seven oral sample types. Heat tree (A) is a labelled key of well-established $NO_3^-$-reducing bacteria, where node size indicates the number of overall reads across all oral sites. The genera on the periphery of heat tree (A) correspond to the nearest branch ending with a dark grey node. Blue and red colours show the $\log_2$-ratio of median proportions observed at each oral site. Blue taxa correspond to the oral site label highlighted in blue, and red taxa correspond to the oral site label highlighted in red. Blue taxa are enriched in the blue oral sites labelled in the row, whilst red taxa are enriched in the red oral sites labelled in the column. The gradient of taxa colours corresponds to the difference in $\log_2$-ratio of bacterial median proportions, as shown in the legend.

## Determining genera in each oral site with the highest percentage median relative abundance

The percentage median relative abundance of genera was calculated for each oral site, and the ten genera found in the highest relative abundances were *Actinomyces, Fusobacterium, Gemella, Granulicatella, Haemophilus, Neisseria, Porphyromonas, Prevotella, Streptococcus,* and *Veillonella (Fig 3)*.

**Table 1. Tables of pairwise comparison differences in log$_2$-ratio median read proportions in selected NO3$^-$-reducing genera at six oral sites and saliva.**

| Genera | Attached keratinized gingiva— Buccal mucosa | Attached Keratinized gingiva— Hard palate | Attached Keratinized gingiva— Saliva | Attached Keratinized gingiva— Subgingival plaque | Attached Keratinized gingiva— Supraginginval plaque | Attached Keratinized gingiva— Tongue dorsum |
|---|---|---|---|---|---|---|
| *Actinomyces* | -3.45 | -4.79 | -4.20 | -5.73 | -6.23 | -5.69 |
| *Brevibacillus* | 0 | 0 | 0 | 0 | 0 | 0 |
| *Campylobacter* | 0 | -2.58 | -5.36 | -4.60 | -3.56 | -3.81 |
| *Capnocytophaga* | -2.44 | -1.51 | -4.18 | -6.82 | -7.07 | -0.95 |
| *Corynebacterium* | -1.00E+25 | -1.00E+25 | -1.00E+25 | -1.00E+25 | -1.00E+25 | 0 |
| *Eikenella* | 0 | 0 | -1.00E+25 | -1.00E+25 | -1.00E+25 | 0 |
| *Fusobacterium* | -1.41 | -1.74 | -3.31 | -4.91 | -3.23 | -3.02 |
| *Granulicatella* | 0 | -1.10 | 0 | 2.01 | 1.62 | -0.98 |
| *Haemophilus* | 0.82 | 1.37 | 0.95 | 3.13 | 2.30 | 0.94 |
| *Leptotrichia* | -3.34 | -3.81 | -4.49 | -5.78 | -6.29 | -4.88 |
| *Microbacterium* | 0 | 0 | 0 | 0 | 0 | 0 |
| *Neisseria* | -2.49 | -3.33 | -3.67 | -2.55 | -3.69 | -4.47 |
| *Porphyromonas* | 0 | -1.05 | -1.70 | -1.22 | 0 | 0 |
| *Prevotella* | 0 | -1.77 | -2.83 | -2.04 | 0 | -2.48 |
| *Propionibacterium* | -1.00E+25 | -1.00E+25 | -1.00E+25 | -1.00E+25 | -1.00E+25 | 0 |
| *Rothia* | -4.37 | -5.33 | -4.53 | -5.83 | -6.82 | -5.36 |
| *Selenomonas* | -1.00E+25 | -1.00E+25 | -1.00E+25 | -1.00E+25 | -1.00E+25 | -1.00E+25 |
| *Staphylococcus* | 0 | 0 | 0 | 0 | 0 | 0 |
| *Streptococcus* | -0.19 | 0 | 1.78 | 2.57 | 1.94 | 1.26 |
| *Veillonella* | 0 | -1.15 | -2.10 | 0 | 0 | -2.17 |
| **Genera** | **Buccal mucosa— Hard palate** | **Buccal mucosa— Saliva** | **Buccal mucosa— Subgingival plaque** | **Buccal mucosa— Supraginginval plaque** | **Buccal mucosa— Tongue dorsum** | |
| *Actinomyces* | -1.34 | -0.75 | -2.28 | -2.79 | -2.25 | |
| *Brevibacillus* | 0 | 0 | 0 | 0 | 0 | |
| *Campylobacter* | -1.93 | -4.71 | -3.94 | -2.91 | -3.16 | |
| *Capnocytophaga* | 0.93 | -1.74 | -4.38 | -4.62 | 1.49 | |
| *Corynebacterium* | 0 | 0 | -5.92 | -6.79 | 1.00E+25 | |
| *Eikenella* | 0 | 0 | -1.00E+25 | -1.00E+25 | 0 | |
| *Fusobacterium* | 0 | -1.90 | -3.51 | -1.82 | -1.62 | |
| *Granulicatella* | -1.37 | 0 | 1.74 | 1.35 | -1.25 | |
| *Haemophilus* | 0.55 | 0 | 2.31 | 1.48 | 0 | |
| *Leptotrichia* | 0 | -1.15 | -2.44 | -2.95 | -1.54 | |
| *Microbacterium* | 0 | 0 | 0 | 0 | 0 | |
| *Neisseria* | -0.84 | -1.18 | 0 | -1.20 | -1.98 | |
| *Porphyromonas* | 0 | -1.22 | -0.74 | 0 | 0 | |
| *Prevotella* | -2.01 | -3.07 | -2.29 | 0 | -2.73 | |
| *Propionibacterium* | 0 | 0 | -4.27 | -4.39 | 1.00E+25 | |
| *Rothia* | 0 | 0 | -1.46 | -2.45 | 0 | |
| *Selenomonas* | -1.41 | -4.73 | -5.40 | -4.61 | 0 | |
| *Staphylococcus* | 0 | 0 | 0 | 0 | 0 | |
| *Streptococcus* | 0.28 | 1.97 | 2.76 | 2.13 | 1.45 | |
| *Veillonella* | -1.33 | -2.28 | 0 | 0 | -2.35 | |
| **Genera** | **Hard palate— Saliva** | **Hard palate— Subgingival plaque** | **Hard palate— Supraginginval plaque** | **Hard palate— Tongue dorsum** | **Saliva— Subgingival plaque** | |
| *Actinomyces* | 0.59 | -0.94 | -1.44 | -0.90 | -1.53 | |
| *Brevibacillus* | 0 | 0 | 0 | 0 | 0 | |

*(Continued)*

**Table 1.** (Continued)

| Genera | Attached keratinized gingiva—Buccal mucosa | Attached Keratinized gingiva—Hard palate | Attached Keratinized gingiva—Saliva | Attached Keratinized gingiva—Subgingival plaque | Attached Keratinized gingiva—Supragingival plaque | Attached Keratinized gingiva—Tongue dorsum |
|---|---|---|---|---|---|---|
| Campylobacter | -2.78 | -2.01 | -0.98 | -1.23 | 0.77 | |
| Capnocytophaga | -2.67 | -5.31 | -5.56 | 0 | -2.64 | |
| Corynebacterium | 0 | -5.99 | -6.86 | 1.00E+25 | -5.36 | |
| Eikenella | -1.00E+25 | -1.00E+25 | -1.00E+25 | 0 | -5.76 | |
| Fusobacterium | -1.57 | -3.18 | -1.49 | -1.29 | -1.60 | |
| Granulicatella | 1.09 | 3.11 | 2.72 | 0 | 2.02 | |
| Haemophilus | -0.42 | 1.76 | 0.93 | -0.43 | 2.18 | |
| Leptotrichia | -0.68 | -1.97 | -2.48 | -1.07 | -1.29 | |
| Microbacterium | 0 | 0 | 0 | 0 | 0 | |
| Neisseria | 0 | 0 | -0.36 | -1.14 | 1.12 | |
| Porphyromonas | -0.66 | 0 | 0 | 0 | 0.48 | |
| Prevotella | -1.06 | 0 | 1.71 | -0.72 | 0.78 | |
| Propionibacterium | 0 | -3.80 | -3.92 | 1.00E+25 | -3.30 | |
| Rothia | 0.80 | 0 | -1.49 | 0 | -1.30 | |
| Selenomonas | -3.32 | -3.99 | -3.20 | 0 | 0 | |
| Staphylococcus | 0 | 0 | 0 | 0 | 0 | |
| Streptococcus | 1.69 | 2.48 | 1.85 | 1.17 | 0.80 | |
| Veillonella | -0.95 | 1.12 | 1.03 | -1.02 | 2.08 | |
| Genera | Saliva—Supragingival plaque | Saliva—Tongue dorsum | Subgingival plaque—Supragingival plaque | Subgingival plaque—Tongue dorsum | Supragingival plaque—Tongue dorsum | |
| Actinomyces | -2.03 | -1.49 | -0.50 | 0 | 0.54 | |
| Brevibacillus | 0 | 0 | 0 | 0 | 0 | |
| Campylobacter | 1.81 | 1.56 | 1.04 | 0.79 | 0 | |
| Capnocytophaga | -2.88 | 3.23 | -0.24 | 5.87 | 6.11 | |
| Corynebacterium | -6.23 | 1.00E+25 | -0.87 | 1.00E+25 | 1.00E+25 | |
| Eikenella | -5.67 | 1.00E+25 | 0 | 1.00E+25 | 1.00E+25 | |
| Fusobacterium | 0 | 0 | 1.69 | 1.89 | 0 | |
| Granulicatella | 1.63 | -0.97 | 0 | -2.99 | -2.60 | |
| Haemophilus | 1.35 | 0 | -0.83 | -2.19 | -1.36 | |
| Leptotrichia | -1.80 | 0 | 0 | 0.90 | 1.41 | |
| Microbacterium | 0 | 0 | 0 | 0 | 0 | |
| Neisseria | 0 | -0.81 | -1.14 | -1.92 | -0.78 | |
| Porphyromonas | 0.87 | 0.96 | 0 | 0 | 0 | |
| Prevotella | 2.77 | 0.34 | 1.99 | 0 | -2.43 | |
| Propionibacterium | -3.43 | 1.00E+25 | 0 | 1.00E+25 | 1.00E+25 | |
| Rothia | -2.29 | -0.83 | -0.99 | 0 | 1.46 | |
| Selenomonas | 0.12 | 4.02 | 0.79 | 4.69 | 3.90 | |
| Staphylococcus | 0 | 0 | 0 | 0 | 0 | |
| Streptococcus | 0 | -0.52 | -0.63 | -1.31 | -0.69 | |
| Veillonella | 1.98 | 0 | 0 | -2.15 | -2.05 | |

*Haemophilus* and *Streptococcus* were found in all oral sites. In four of the seven oral sample types, *Streptococcus* was the most abundant (attached keratinized gingiva, 47%; buccal mucosa, 54%; hard palate, 45%). *Prevotella* and *Streptococcus* were found at the highest median relative abundance in saliva (15%). Saliva had a higher abundance of "Other" bacteria, which included

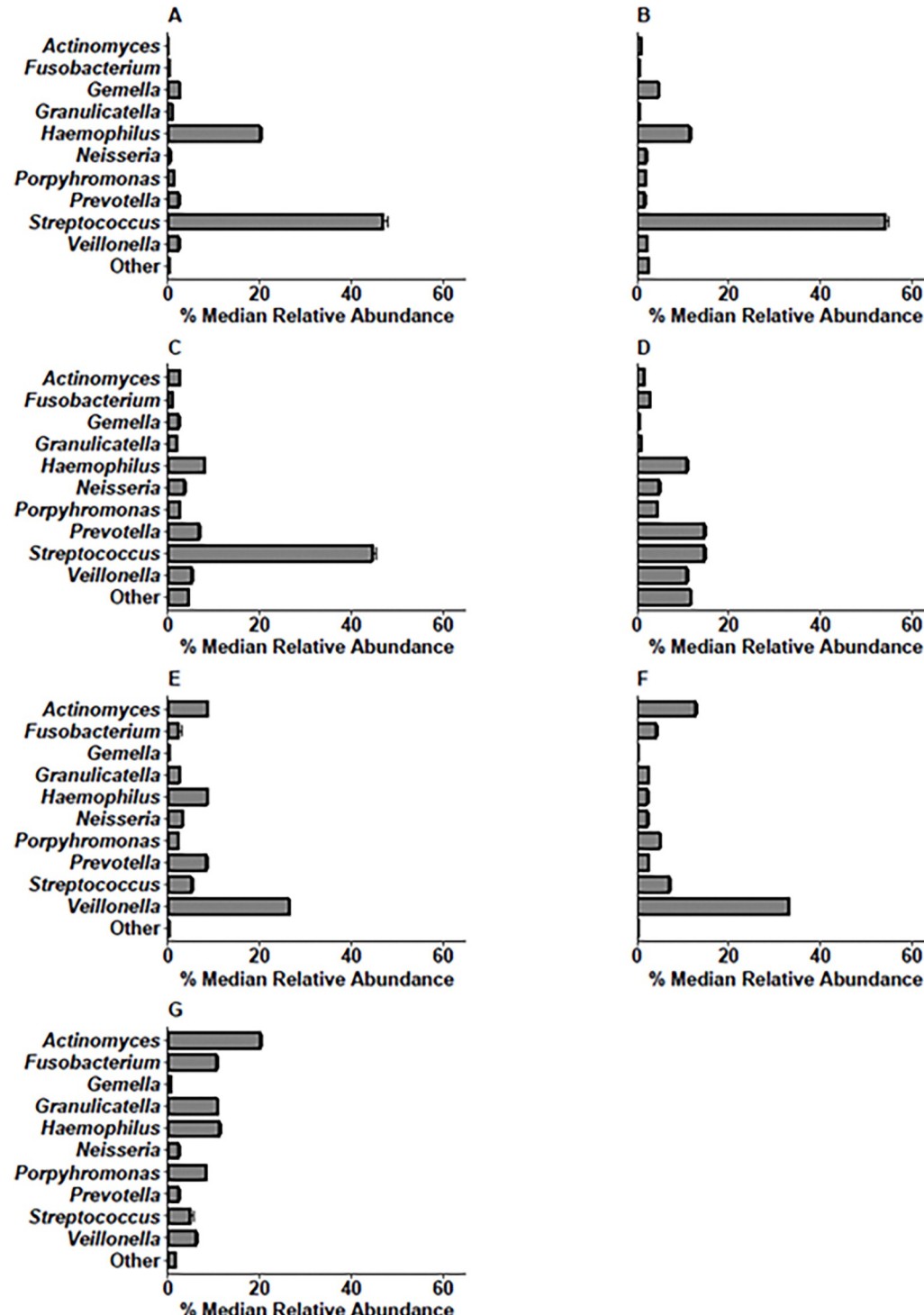

**Fig 3. Bar charts showing the percentage median relative abundance of bacterial genera in the attached keratinized gingiva, buccal mucosa, hard palate, saliva, subgingival plaque, supragingival plaque and tongue dorsum. A** attached keratinized gingiva, **B** buccal mucosa, **C** hard palate, **D** saliva, **E** subgingival plaque, **F** supragingival plaque, **G** tongue dorsum. Error bars show the standard error of median. The "Other" bacteria refer to the sum of the genera observed at a lower abundance.

the combined genera of lower relative abundances (12%). *Veillonella* were found in a higher median relative abundance in the saliva (11%), subgingival plaque (26%) and supragingival plaque (33%). In the tongue dorsum, subgingival and supragingival plaque sites, there was also a higher percentage median relative abundance of *Actinomyces* compared to other sites.

### Visual analysis of clustering using multidimensional scaling plots

Multidimensional scaling plots of each of the seven oral sample types showed that there was low intra-site variability, and higher inter-site variability compared to intra-site variability (*Fig 4*). Distinct clustering was found in the hard palate, subgingival plaque, supragingival plaque, and tongue dorsum sites. The attached keratinized gingiva and buccal mucosa sites visually showed less pronounced clustering of individual samples. Saliva clustering of individual samples was the least apparent amongst the different oral sample types, with there being further distances between individual samples.

## Discussion

The relationships between the abundance of oral bacteria involved in the $NO_3^-$-$NO_2^-$-NO pathway, NO homeostasis and physiological processes, have been recognised as a potential target for improving blood pressure regulation, cognition, protection against ischaemia-reperfusion injury and exercise performance [3, 6–9]. Saliva and tongue swab samples are frequently used as measures of the oral microbial community and may be particularly useful for studies which aim to relate microbial communities to physiological or clinical outcomes. Saliva and tongue swab sampling may be used due to ease of sampling and high bacterial diversity and density [16–19]. However, due to discrepancies between sampling methods in previous reports, including whether the saliva or tongue dorsum was sampled, it is important to establish whether higher proportions of $NO_3^-$-reducing bacteria were located elsewhere in the oral cavity. Our results provide an insight into which oral niches may be most appropriate to sample $NO_3^-$-reducing bacteria involved in the $NO_3^-$-$NO_2^-$-NO pathway and highlight the benefits of saliva sampling in future physiological studies.

The overall oral bacterial composition was consistent with previous literature, where Firmicutes, Actinobacteria, Bacteroidetes, Proteobacteria and Fusobacteria were the most abundant phyla [25, 26]. This is further reflected at the family and genus levels, with *Streptococcus*, *Prevotella*, *Neisseria* and *Haemophilus* being most abundant genera [1, 15, 16, 26].

A limitation to the present study was that detailed participant clinical information was not available on the publicly accessible HMP database. Therefore, MDS plots were generated to qualitatively establish any outliers within each oral site. Further research is warranted to compare oral microbiomes related to specific host characteristics, including sex and health status. As clustering was generally observed in six of the seven oral sample types, samples were not immediately excluded, as the distances between each point may not reflect actual dissimilarity within each site. Therefore, the MDS plots overall showed low intra-individual variability in most oral sites. Saliva displayed higher intra-individual variability, represented by the spread between data points. However, because clinical information was not available, it is unclear whether this was due to differences between participants in the HMP cohort, or because of the high diversity of bacteria typically found in saliva [26, 27]. Furthermore, it is important to note that variation between samples of the same site may also occur due to differences in sampling techniques, for example, the pressure used to sample bacteria caught within the tongue crypt. Further understanding of how sampling techniques could influence the recording of bacterial abundance may be useful for future studies [28].

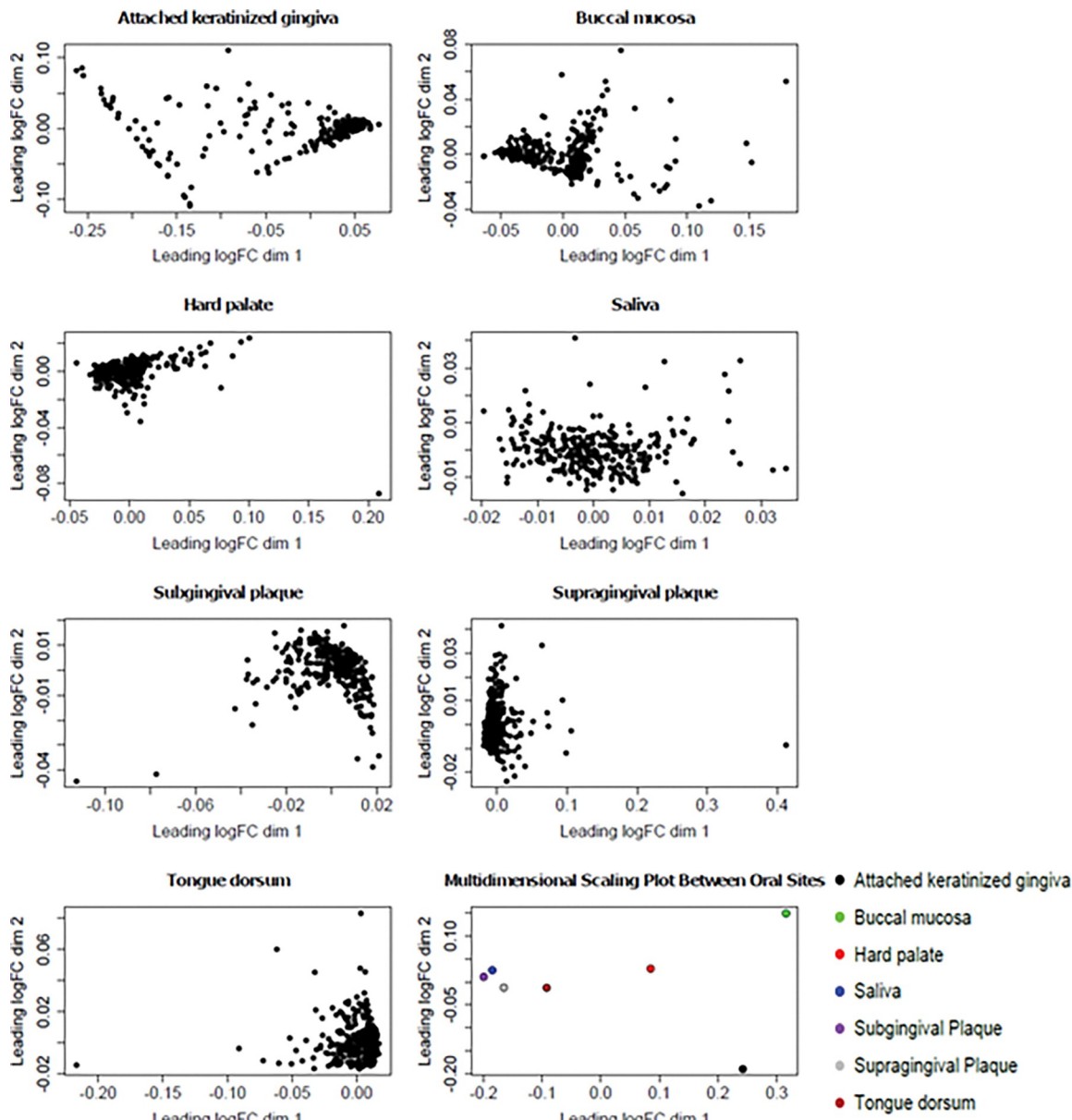

**Fig 4. Multidimensional scaling plots (MDS) of individual samples in the attached keratinized gingiva, buccal mucosa, hard palate, saliva, subgingival plaque, supragingival plaque and tongue dorsum.** The multidimensional scaling plot of oral sites shows that the samples taken at each site share similarities. The attached keratinized gingiva shows some clustering, although there is higher intra-variability than the other sites, excluding saliva. Buccal mucosa samples appear to form two clusters which are not present within the other samples. The hard palate, subgingival plaque, supragingival plaque and tongue dorsum show clustering. Saliva samples appear to have the most variability between points.

## Comparison of alpha diversity at each oral site

The high diversity and species richness found in saliva and plaque sites compared to the other oral sites represents a larger mixture of microbial communities, which may also account for the higher intra-individual variability compared to the diversity and species richness found in the other oral sites [26, 27]. The high bacterial diversity, species richness and variability in saliva samples may be because saliva samples represent a composite of bacteria from all oral

sites. Furthermore, variability in the salivary microbiome between humans may be a result of oral hygiene routine or physiological factors such as obesity and fitness [26, 27, 29].

## Pairwise comparisons of predominant and $NO_3^-$-reducing genera $\log_2$-ratio of median proportions at saliva and six oral sites

Pairwise comparisons between each oral site were used to determine which sites had the highest $\log_2$-ratio of median proportion of 20 predominant and highly abundant $NO_3^-$-reducing bacteria. The well-established predominant and highly abundant $NO_3^-$-reducing bacteria were selected based on previous studies [13, 16], and included *Actinomyces*, *Brevibacillus*, *Campylobacter*, *Capnocytophaga*, *Corynebacterium*, *Eikenella*, *Fusobacterium*, *Granulicatella*, *Haemophilus*, *Leptotrichia*, *Microbacterium*, *Neisseria*, *Porphyromonas*, *Prevotella*, *Propionibacterium*, *Rothia*, *Selenomonas*, *Staphylococcus*, *Streptococcus*, and *Veillonella*.

Saliva had significantly enriched proportions of eight $NO_3^-$-reducing bacteria, whilst the tongue dorsum had a significantly higher proportion of *Actinomyces*, *Granulicatella*, *Neisseria*, *Rothia*, and *Streptococcus*, confirming the results of Aas *et al*. [1]. These results are important, as the tongue dorsum has been suggested as the site where most $NO_3^-$-reduction occurs [13], but a higher abundance of $NO_3^-$-reducing bacteria could be sampled from other oral sample types such as the saliva, which may be more appropriate for physiological studies that aim to collect samples of oral $NO_3^-$-reducing bacteria.

In the attached keratinized gingiva site, the selected $NO_3^-$-reducing genera had significantly reduced median proportions compared to the other oral sites. However, in all of the pairwise comparisons, *Haemophilus* was significantly enriched in the attached keratinized gingiva site, followed by *Streptococcus* which was significantly enriched when compared to the saliva, plaque sites and the tongue dorsum. The hard palate also had significantly enriched proportions of *Streptococcus* This may be a result of high bacterial cocci colonisation on the hard oral surfaces, such as *Streptococcus* and *Haemophilus*, where cocci bacteria can efficiently attach to epithelial cells and tooth enamel [30]. A similar result was found for the buccal mucosa site, consistent with previous studies [1, 16].

The subgingival and supragingival plaque sites exhibited less notable differences, consistent with previous research [31, 32]. Supragingival plaque had a significantly higher abundance of predominant $NO_3^-$-reducing bacteria compared to subgingival plaque. The higher proportions of $NO_3^-$-reducing bacteria in supragingival plaque are likely due to positioning within the oral cavity. Supragingival plaque is in frequent contact with saliva, and may also be more exposed to bacteria of the outside environment, such as in air or water, whereas subgingival plaque is located deeper in the oral cavity [31].

## Percentage median relative abundances of genera in each oral site

The tongue dorsum had a low percentage median relative abundance of "Other" bacteria, which may be a result of the tongue scrapings reaching deeper oral niches in the tongue surface. The tongue dorsum also had a higher percentage median relative abundance of *Actinomyces* compared to the other oral sites. This may be because the tongue dorsum crypts are protected from the sheer force of salivary flow [27]. In saliva, the frequently recognised $NO_3^-$-reducing *Prevotella* were found to be one of the genera with the highest percentage mean relative abundance. Consistent with the alpha diversity analyses, saliva had a high percentage median relative abundance of "Other" bacteria, compared to the specific oral sites. This may be because saliva represents $NO_3^-$-reducing bacteria accumulated from all oral sites, as well as from food and fluid consumption. High proportions of *Streptococcus*, found within the oral sites are consistent with previous findings [14]. After undertaking routine oral hygiene

procedures, *Streptococcus*, are typically some of the first to colonise oral sites, with these bacteria utilising facilitated attachment by salivary glycoproteins [30, 31]. High counts of *Streptococcus* may also account for a high abundance of *Veillonella*, and *Actinomyces*, as these bacteria form a symbiotic relationship within oral biofilms [32, 33].

## Conclusions

Comparisons of the oral $NO_3^-$-reducing genera in different oral sample types, using the HMP dataset, revealed the importance of selecting an appropriate oral site from which to sample bacteria likely to be involved in the $NO_3^-$-$NO_2^-$-NO pathway. Using only one method of sampling oral bacteria may lead to low abundances of $NO_3^-$-reducing genera, particularly in small population physiological studies. Saliva samples are likely to provide a good representation of all $NO_3^-$-reducing bacteria in the oral cavity, providing a high yield of $NO_3^-$-reducing bacteria samples when compared to the tongue dorsum sample site. However, the measurement of bacterial abundance using saliva samples alone may lead to more variability between samples of the same type and may not capture bacteria deep within biofilms found in the plaque sites or crevices of the tongue dorsum. This comparison of $NO_3^-$-reducing genera at each oral site provides a basis for where to sample $NO_3^-$-reducing bacteria in future physiological studies investigating the relationship between the bacteria in the $NO_3^-$-$NO_2^-$-NO pathway and human health.

## Supporting information

**S1 File.**
(ZIP)

## Acknowledgments

We thank Dr Ryan Ames for his valuable advice on data analysis during this project. We thank the Human Microbiome Project for the publicly available data used for analysis in this study.

## Author Contributions

**Conceptualization:** Joanna E. L'Heureux, Mark van der Giezen, Paul G. Winyard, Andrew M. Jones, Anni Vanhatalo.

**Formal analysis:** Joanna E. L'Heureux.

**Methodology:** Joanna E. L'Heureux.

**Software:** Joanna E. L'Heureux.

**Supervision:** Mark van der Giezen, Paul G. Winyard, Andrew M. Jones, Anni Vanhatalo.

**Writing – original draft:** Joanna E. L'Heureux.

**Writing – review & editing:** Mark van der Giezen, Paul G. Winyard, Andrew M. Jones, Anni Vanhatalo.

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
