## [Decision Letter · Decision Letter 0]

7 Sep 2023

PONE-D-23-17952Localisation of nitrate-reducing and highly abundant microbial communities in the oral cavityPLOS ONE

Dear Dr. L'Heureux,

Thank you for submitting your manuscript to PLOS ONE. After careful consideration, we feel that it has merit but does not fully meet PLOS ONE’s publication criteria as it currently stands. Therefore, we invite you to submit a revised version of the manuscript that addresses the points raised during the review process.

We look forward to receiving your revised manuscript.

Kind regards,

Artak Heboyan, Ph.D.

Academic Editor

PLOS ONE

Journal Requirements:

Reviewers' comments:

Reviewer's Responses to Questions

**Comments to the Author**

1. Is the manuscript technically sound, and do the data support the conclusions?

Reviewer #1: Yes

Reviewer #2: Yes

Reviewer #3: No

2. Has the statistical analysis been performed appropriately and rigorously? 

Reviewer #1: Yes

Reviewer #2: Yes

Reviewer #3: No

3. Have the authors made all data underlying the findings in their manuscript fully available?

Reviewer #1: Yes

Reviewer #2: Yes

Reviewer #3: Yes

4. Is the manuscript presented in an intelligible fashion and written in standard English?

Reviewer #1: Yes

Reviewer #2: Yes

Reviewer #3: Yes

5. Review Comments to the Author

Reviewer #1: The manuscript describes the localization of nitrate-reducing and microbial communities in the oral sites such as attached gingiva, buccal mucosa, hard palate, saliva, supra- & subgingival plaque, and tongue dorsum. The contents (data) of the manuscript seem suitable for the publication of the PLoS One. The specific points are as follows;

[Suggestions]

>> "Resuls" and Figure 3:

(P. 14, L. 236-237)

"Haemophilus sp. and Streptococcus sp. were found in high percentage median relative abundance in all oral sites."

The referee feels that "Haemophilus sp. in the supragingival plaque" and "Streptococcus sp.in the tongue dorsum" were not so high in the Figure 3.

>> "Discussiom"

(P. 17, L. 301-305)

"Furthermore, it is important to note that variation between samples of the same site may also occur due to differences in sampling techniques, for example, the pressure used to sample bacteria caught within the tongue crypt. Further understanding of how sampling techniques could influence the recording of bacterial abundance may be useful for future studies."

For the authors;

The referee has an experience of reading literature as follows,

1) Kikutani et al: A novel rapid oral bacteria detection apparatus for effective oral care to prevent pneumonia. Gerodontology 2012; 29(2): e560-e565), it says "The collection pressure was about 20 g, and a 1-cm distance was rubbed back and forth three times with a swab." And additionally,

2) Sato-Suzuki et al: Nitrite-producing oral microbiome in adults and children. Sci Rep 10: 16652 (11 pages), 2020.

Reviewer #2: I have reviewed the manuscript entitled “Localisation of nitrate-reducing and highly abundant microbial communities in the oral cavity” submitted for possible publication in the journal “PLOS ONE”. The authors have done great efforts in compiling the results and describing the material and methods used in their study. The topic of manuscript is of interest of the reader and belongs to one of the most emerging issues in healthcare systems. The paper can significantly contribute to the field. As the authors have described very well, I didn’t notice any major flaw or suggestions for further improvement in the manuscript. Although some minor changes have to be done before proceeding it further procedure. My specific comments are:

1. I suggest to write full forms for NO etc at it’s first appearance in the abstract and then at introduction.

2. Line 16-17: The sentence needs to be rephrased.

3. Abstract: Write down the aims of study.

4. Line 55: change as “The Human Microbiome Project (HMP) under National Institute of Health (NIH)

5. Line 63-68: These could be better representative for the discussion section.

6. Introduction: At the end of this section, mention about the study rationale and objectives.

7. Line 88: change “NIH Human Microbiome Project” to NIH’s HMP”.

8. Line 137-144: Mention here about the p value, what p value was considered as significant? what parameter they used to calculate the p value.

9. Line 188: The authors are suggested to write “spp.” Instead of “sp.”

10. Line 239 and others: remove space between number and %.

Reviewer #3: In the present study, L'Heureux et. al. and colleagues investigated the localisation of nitrate-reducing and highly abundant microbial communities in the oral cavity. The topic explored in this article is interesting, however, there are several points in this manuscript that need to be addressed in more detail and are listed below.

1. the authors did not provide clinical information on the study participants, e.g. healthy/disease status. Several systemic diseases are known to affect the oral microbiome and lead to dysbiosis. Please discuss this point in more details

2. The authors mention several well-established NO3-reducing bacteria, but not all reduce NO3 with the same efficiency. Please provide more information about the bacterial taxa with the highest NO3 reducing power and rank the taxa by their efficiency from top to bottom.

3.Please add a new paragraph explaining the limitations of the study in more detail.

4.In Figure 2, please italicize all bacterial names.

5. Did the authors examine the differences between males and females for each sample? Please create a new MDS figure showing the sex differences within the samples.

6. PLOS authors have the option to publish the peer review history of their article (what does this mean?). If published, this will include your full peer review and any attached files.

Reviewer #1: No

Reviewer #2: No

Reviewer #3: **Yes: **Mohamed Abdelbary

---

## [Author Response · Author response to Decision Letter 0]

3 Nov 2023

RESPONSE: Thank you for providing the style requirements. We have revised the manuscript accordingly.

RESPONSE: The data are currently publicly available in the Human Microbiome Project repository that is referenced in the manuscript. The data will be available with a DOI provided through the University of Exeter ORE once the article has been accepted for publication.

 

Reviewers' comments:

Reviewer's Responses to Questions

Comments to the Author

1. Is the manuscript technically sound, and do the data support the conclusions?

Reviewer #1: Yes

Reviewer #2: Yes

Reviewer #3: No

2. Has the statistical analysis been performed appropriately and rigorously? 

Reviewer #1: Yes

Reviewer #2: Yes

Reviewer #3: No

5. Review Comments to the Author

Reviewer #1: The manuscript describes the localization of nitrate-reducing and microbial communities in the oral sites such as attached gingiva, buccal mucosa, hard palate, saliva, supra- & subgingival plaque, and tongue dorsum. The contents (data) of the manuscript seem suitable for the publication of the PLoS One. The specific points are as follows;

RESPONSE: We thank the reviewer for helpful comments that have improved the manuscripts.

[Suggestions]

>> "Resuls" and Figure 3:

(P. 14, L. 236-237)

"Haemophilus sp. and Streptococcus sp. were found in high percentage median relative abundance in all oral sites."

The referee feels that "Haemophilus sp. in the supragingival plaque" and "Streptococcus sp.in the tongue dorsum" were not so high in the Figure 3.

RESPONSE: Lines 236-237 have been revised accordingly (‘Haemophilus sp. and Streptococcus sp. were found in all oral sites’).

>> "Discussiom"

(P. 17, L. 301-305)

"Furthermore, it is important to note that variation between samples of the same site may also occur due to differences in sampling techniques, for example, the pressure used to sample bacteria caught within the tongue crypt. Further understanding of how sampling techniques could influence the recording of bacterial abundance may be useful for future studies."

For the authors;

The referee has an experience of reading literature as follows,

1) Kikutani et al: A novel rapid oral bacteria detection apparatus for effective oral care to prevent pneumonia. Gerodontology 2012; 29(2): e560-e565), it says "The collection pressure was about 20 g, and a 1-cm distance was rubbed back and forth three times with a swab." And additionally,

2) Sato-Suzuki et al: Nitrite-producing oral microbiome in adults and children. Sci Rep 10: 16652 (11 pages), 2020.

RESPONSE: Thank you for drawing our attention to these articles that support our statement. We have revised referencing accordingly.

Reviewer #2: I have reviewed the manuscript entitled “Localisation of nitrate-reducing and highly abundant microbial communities in the oral cavity” submitted for possible publication in the journal “PLOS ONE”. The authors have done great efforts in compiling the results and describing the material and methods used in their study. The topic of manuscript is of interest of the reader and belongs to one of the most emerging issues in healthcare systems. The paper can significantly contribute to the field. As the authors have described very well, I didn’t notice any major flaw or suggestions for further improvement in the manuscript. Although some minor changes have to be done before proceeding it further rocedure. My specific comments are:

RESPONSE: We thank the reviewer for a constructive and positive review of our submission.

1. I suggest to write full forms for NO etc at it’s first appearance in the abstract and then at introduction.

RESPONSE: This has been corrected in the abstract and introduction.

2. Line 16-17: The sentence needs to be rephrased.

RESPONSE: This sentence now reads “NO3--reducing oral bacteria reduce inorganic dietary NO3- to nitrite (NO2-) via the NO3--NO2--NO pathway.”

3. Abstract: Write down the aims of study.

RESPONSE: We have revised the abstract to include the study aim (“The aim of this study was to assess whether other areas in the mouth could contain a physiologically more relevant abundance of NO3- reducing bacteria, which may be important for sampling in clinical studies.”)

4. Line 55: change as “The Human Microbiome Project (HMP) under National Institute of Health (NIH)

RESPONSE: This has been revised as suggested.

5. Line 63-68: These could be better representative for the discussion section.

RESPONSE: Thank you for this suggestion. However, we feel that this section forms a key part of the study rationale and ask the reviewer’s discretion to maintain it in the introduction.

6. Introduction: At the end of this section, mention about the study rationale and objectives.

RESPONSE: We have revised lines 80-85 accordingly. (“The aim of this study was, therefore, to determine the site(s) in the mouth where the highest abundance of NO3--reducing bacteria reside in a larger human population, such as those found in the HMP, in order to inform methodology for future studies investigating the relationships between oral NO3--reducing bacteria and human health.”)

7. Line 88: change “NIH Human Microbiome Project” to NIH’s HMP”.

RESPONSE: We do not feel that the use of genitive case is necessary here and ask the reviewer’s discretion in maintaining the commonly used terminology (e.g. https://www.ncbi.nlm.nih.gov/pmc/articles/PMC2792171/).

8. Line 137-144: Mention here about the p value, what p value was considered as significant? what parameter they used to calculate the p value.

RESPONSE: We have clarified that P<0.05 was considered statistically significant in line 147.

9. Line 188: The authors are suggested to write “spp.” Instead of “sp.”

RESPONSE: Thank you. We have remover “sp.” from this narrative given that only genus level data were presented for analysis.

10. Line 239 and others: remove space between number and %.

RESPONSE: This has been revised throughout the manuscript.

Reviewer #3: In the present study, L'Heureux et. al. and colleagues investigated the localisation of nitrate-reducing and highly abundant microbial communities in the oral cavity. The topic explored in this article is interesting, however, there are several points in this manuscript that need to be addressed in more detail and are listed below.

1.The authors did not provide clinical information on the study participants, e.g. healthy/disease status. Several systemic diseases are known to affect the oral microbiome and lead to dysbiosis. Please discuss this point in more details

RESPONSE: The HMP participants were healthy 18- to 40-yr-old adults, who were screened using exclusion criteria based on health history, including the presence of systemic diseases (e.g., hypertension, cancer, or immunodeficiency or autoimmune disorders), use of potential immunomodulators, and recent use of antibiotics or probiotics. We have added this information in the methods section (line 96-100).

2. The authors mention several well-established NO3-reducing bacteria, but not all reduce NO3 with the same efficiency. Please provide more information about the bacterial taxa with the highest NO3 reducing power and rank the taxa by their efficiency from top to bottom.

RESPONSE: We thank the reviewer for raising this consideration and we have added reference to previous research (Doel et al. 2005, Hyde et al. 2014) that has assessed efficiency of given species for nitrate reduction in vitro (lines 74-81).

3.Please add a new paragraph explaining the limitations of the study in more detail.

RESPONSE: We have revised the paragraph highlighting study limitations as suggested (line 305-320).

4.In Figure 2, please italicize all bacterial names.

RESPONSE: This genera names have been italicized.

5. Did the authors examine the differences between males and females for each sample? Please create a new MDS figure showing the sex differences within the samples.

RESPONSE: We checked for sex differences within each oral site before proceeding with further analysis. There were no sex differences, therefore, the male and female samples were grouped together. This information has been added in lines 130-134.

 

6. PLOS authors have the option to publish the peer review history of their article (what does this mean?). If published, this will include your full peer review and any attached files.

Do you want your identity to be public for this peer review? For information about this choice, including consent withdrawal, please see our Privacy Policy.

Reviewer #1: No

Reviewer #2: No

Reviewer #3: Yes: Mohamed Abdelbary

---

## [Decision Letter · Decision Letter 1]

15 Nov 2023

Localisation of nitrate-reducing and highly abundant microbial communities in the oral cavity

PONE-D-23-17952R1

Dear Dr. L'Heureux,

We’re pleased to inform you that your manuscript has been judged scientifically suitable for publication and will be formally accepted for publication once it meets all outstanding technical requirements.

Kind regards,

Artak Heboyan, Ph.D.

Academic Editor

PLOS ONE

Additional Editor Comments (optional):

Reviewers' comments:

Reviewer's Responses to Questions

**Comments to the Author**

1. If the authors have adequately addressed your comments raised in a previous round of review and you feel that this manuscript is now acceptable for publication, you may indicate that here to bypass the “Comments to the Author” section, enter your conflict of interest statement in the “Confidential to Editor” section, and submit your "Accept" recommendation.

Reviewer #1: All comments have been addressed

Reviewer #2: All comments have been addressed

2. Is the manuscript technically sound, and do the data support the conclusions?

Reviewer #1: Yes

Reviewer #2: Yes

3. Has the statistical analysis been performed appropriately and rigorously? 

Reviewer #1: Yes

Reviewer #2: Yes

4. Have the authors made all data underlying the findings in their manuscript fully available?

Reviewer #1: Yes

Reviewer #2: Yes

5. Is the manuscript presented in an intelligible fashion and written in standard English?

Reviewer #1: Yes

Reviewer #2: Yes

6. Review Comments to the Author

Reviewer #1: (No Response)

Reviewer #2: (No Response)

7. PLOS authors have the option to publish the peer review history of their article (what does this mean?). If published, this will include your full peer review and any attached files.

Reviewer #1: No

Reviewer #2: **Yes: **Naveed Ahmed

---

## [Editor Report · Acceptance letter]

13 Dec 2023

PONE-D-23-17952R1 

PLOS ONE

Dear Dr. L'Heureux, 

I'm pleased to inform you that your manuscript has been deemed suitable for publication in PLOS ONE. Congratulations! Your manuscript is now being handed over to our production team.

Kind regards, 

on behalf of

Dr. Artak Heboyan 

Academic Editor

PLOS ONE